# Improving the Sustainability of Catalytic Glycolysis of Complex PET Waste through Bio-Solvolysis

**DOI:** 10.3390/polym16010142

**Published:** 2024-01-02

**Authors:** Izotz Amundarain, Sheila López-Montenegro, Laura Fulgencio-Medrano, Jon Leivar, Ana Iruskieta, Asier Asueta, Rafael Miguel-Fernández, Sixto Arnaiz, Beñat Pereda-Ayo

**Affiliations:** 1GAIKER Technology Centre, Basque Research and Technology Alliance (BRTA), Parque Tecnológico de Bizkaia, Edificio 202, 48170 Zamudio, Spainasueta@gaiker.es (A.A.); arnaiz@gaiker.es (S.A.); 2Department of Chemical Engineering, University of the Basque Country (UPV/EHU), Barrio Sarriena s/n, 48940 Leioa, Spainbenat.pereda@ehu.eus (B.P.-A.)

**Keywords:** poly(ethylene terephthalate) waste, chemical recycling, glycolysis, biobased, monomer, purification

## Abstract

This work addresses a novel bio-solvolysis process for the treatment of complex poly(ethylene terephthalate) (PET) waste using a biobased monoethylene glycol (BioMEG) as a depolymerization agent in order to achieve a more sustainable chemical recycling process. Five difficult-to-recycle PET waste streams, including multilayer trays, coloured bottles and postconsumer textiles, were selected for the study. After characterization and conditioning of the samples, an evaluation of the proposed bio-solvolysis process was carried out by monitoring the reaction over time to determine the degree of PET conversion (91.3–97.1%) and bis(2-hydroxyethyl) terephthalate (BHET) monomer yield (71.5–76.3%). A monomer purification process, using activated carbon (AC), was also developed to remove the colour and to reduce the metal content of the solid. By applying this purification strategy, the whiteness (L*) of the BHET greatly increased from around 60 to over 95 (L* = 100 for pure white) and the Zn content was significantly reduced from around 200 to 2 mg/kg. The chemical structure of the purified monomers was analyzed via infrared spectroscopy (FTIR) and differential scanning calorimetry (DSC), and the composition of the samples was measured by proton nuclear magnetic resonance (^1^H-NMR), proving a high purity of the monomers with a BHET content up to 99.5% in mol.

## 1. Introduction

Plastics have become essential in the context of human development due to their low cost of production, durability, good mechanical properties and constant growth of usage in many different sectors, such as packaging, building, automotive, electronics and medicine. In 2021, the estimated global production of plastics reached 390.7 million tons, of which Europe produced 57.2 million tons [1]. This high plastic production leads to the generation of a large amount of waste since many plastic products, as packages, have a very short usage life. To handle the increasing quantities of generated plastic waste, the current plastic waste management comprises landfill disposal, recycling and energy recovery alternatives. In 2020, 29.5 million tons of post-consumer plastic waste were sorted and collected in Europe. Of these, 42.0% were incinerated, 34.6% were recycled and 23.4% were landfilled [2]. Today, post-consumer plastic waste recycling is mainly based on mechanical recycling technologies [3]. Mechanical recycling is defined as a sequence of operations to separate, clean, melt and extrude the material to form new plastic parts [4]. Due to the degradation of the virgin polymer and additives, the plastic loses its properties after each cycle, which results in a poorer-quality plastic material after some cycles [5]. Consequently, the material suffers a downcycling and the potential treatment method is incineration or landfill storage [6]. Chemical recycling is presented as the solution to this problem, being a set of technologies complementary to traditional mechanical recycling. By means of chemical recycling, polymeric materials can be depolymerized into monomers or precursors for those monomers that can be used in further polymerization of new plastics. For instance, poly(ethylene terephthalate) (PET) can be chemically recycled by means of glycolysis [7] or hydrolysis [8] to produce bis(2-hydroxyethyl) terephthalate (BHET) or terephthalic acid (TPA), respectively [9]. Other plastics such as polyurethane (PU) foams and end-of-life polyamide (PA) can also be depolymerized in order to obtain polyols [10,11] or caprolactam [12], respectively.

PET is a polymeric material widely used in packaging applications, including sheet and bottle production, due to its excellent properties such as light weight, transparency and high mechanical strength [13]. In 2020, it was estimated that 4.6 million tons of rigid PET packaging material reached the end of its useful life and were, therefore, available for collection across the EU27 + UK. The large majority of this was PET bottles (3.6 million tons), the remainder being PET trays (1.0 million tons) [14]. Of these, just 2.4 million tons (52%) were in fact collected and recycled. Of the rest, most of the waste ends up in incineration plants or landfill sites, and a significant amount ends up in the sea [15]. In practice, the mechanical recycling method is commonly used to recycle waste PET. However, even though PET can be reprocessed at low temperatures, the reprocessing of PET at a high temperature deteriorates or alters the properties of the recycled products [16]. Recently, several chemical recycling technologies have been studied due to their high potential regarding the improvement of circularity in PET waste recycling [17]. Among the different chemical recycling processes for PET materials, depolymerization into monomeric compounds can be achieved via solvolysis, where a solvent, such as ethylene glycol (EG), methanol, water, ammonia or amines, is used as a degradation agent [18]. Glycolysis is one of the most researched depolymerization processes for PET waste, due to the use of reduced quantities of reagents, as well as the use of a low operating temperature and pressure compared to other methods such as methanolysis, while hydrolysis under acid or basic conditions entails serious problems of corrosion and management of the effluents produced. Glycolysis of PET usually uses EG as a degradation agent. The PET reacts with the glycol breaking the ester bonds of the polymer chain and replacing them with hydroxyl groups, obtaining different oligomers and finally the BHET monomer [19]. The reaction rate of glycolysis depends on a number of parameters including temperature, pressure, PET/EG ratio and the type and amount of catalyst. The most widely studied catalysts for glycolysis of PET are zinc acetate and manganese acetate [20,21,22,23,24,25,26,27,28]. Nevertheless, zinc has been reported as the cation with the highest catalytic effect among the corresponding metal acetates for PET glycolysis [29]. Also, the conversion of dimer to a BHET monomer is a reversible process. Therefore, extending the reaction after equilibrium has been reached will cause the reaction to shift backwards, increasing the amount of dimer with the loss of the BHET monomer [30].

Nevertheless, glycolysis (Figure 1) has been reported as one of the best options to chemically recycle PET streams due to the relatively low volatility of the diol solvent, the feasibility of continuous operation and its adaptability to the existing polymerization processes at an industrial scale [31]. In fact, pilot plants treating PET waste through chemical recycling in Europe are at various stages of development. These plants, considered to reach technology readiness levels (TRL) of either “the system is complete and qualified” or there is an “actual system proven in operational environment”, are estimated to have a current input capacity of 68 kt per annum of prepared post-consumer PET flakes [32]. Among these plants, leading companies such as Ioniqa, Garbo, CuRe Technology, Jeplan or Nan Ya Plastics have made significant advances in the upscaling of the glycolysis process for the chemical recycling of post-consumer PET waste [33].

As previously discussed, the lack of appropriate sorting and recycling technologies makes full recycling of PET bottle and tray waste streams difficult. Furthermore, due to the variability and complexity of this type of product (colour and multilayer composition) its mechanical recycling is usually ineffective [34]. Although previous research has identified glycolysis as one of the most suitable chemical recycling processes for PET recycling, applications developed from these findings mainly focus on clean and transparent PET flakes with known composition [35]. Therefore, throughout this work chemical depolymerization via glycolysis will be explored for material valorization of complex PET waste samples, since it allows us to eliminate the impurities present in these streams and to produce the PET starting monomer again [36].

Recent literature on PET solvolysis also indicates that efforts have been made to proceed glycolysis efficiently under mild reaction conditions [37]. Use of a catalyst is a common and popular way to achieve lower reaction temperatures, and the consulted literature mainly focuses on the use of heterogeneous organocatalysts [38,39]. Glycolysis of PET can also be carried out in the presence of a co-solvent in order to improve the performance of the reaction [40]. Although previous research has identified glycolysis as one of the most suitable chemical recycling processes for PET recycling, applications developed from these findings mainly focus on relatively clean PET materials with a known composition. Furthermore, the present work deals with real PET waste that is currently being generated and landfilled or incinerated in large quantities, so this paper aims to present a technically feasible solution based on the principles of the circular economy, thus allowing the manufacture of new value-added products and closing the cycle of PET materials through the application of a chemical recycling process. Nevertheless, the consulted literature suggests that no biobased glycol has been studied for glycolysis of PET until now. Therefore, this work aims to explore the feasibility of a biobased monoethylene glycol (BioMEG from UPM Biochemicals, Helsinki, Finland) for the glycolysis of complex PET waste and study the progress of the reaction by means of HPLC and NMR. The monomers obtained were further purified and analyzed in order to check if the specifications for high purity applications such as food contact are met.

## 2. Experimental

### 2.1. Materials

All PET waste samples were provided by the company Sacyr Environment (Madrid, Spain). The samples received at Gaiker consist of real PET waste that present different degrees of complexity and therefore are not treated by mechanical recycling methods (Table 1). Among these PET samples, multilayer trays, coloured bottles, and polyester textile from both postconsumer clothing and end-of-life tyres can be found (Figure 2). All samples were characterized and conditioned by means of a blade mill to particle sizes smaller than 6 mm in diameter prior to the glycolysis reaction. Biobased monoethylene glycol (BioMEG) was kindly provided by UPM Biochemicals (Helsinki, Finland). This glycol is produced from carbon-neutral feedstocks such as hardwood, which is taken from sustainably managed, certified forests and produced on land not competing with agricultural uses. Zinc acetate (ZnAc_2_) catalyst and reference BHET were supplied by Sigma Aldrich-Merck (Darmstadt, Germany). NORIT^®^ KB-EV SUPRA activated carbon (AC) was provided by Cabot (Boston, MA, USA).

### 2.2. Characterization of PET Waste

Since moisture can interfere with the characterization of PET or could also cause hydrolytic degradation during processing resulting in a reduction in the molecular weight [41], all samples were conditioned for 12 h at 105 °C in an oven before any characterization or treatment. The density and moisture content of PET samples were determined by gravimetric analysis. The chemical structure of the PET samples was measured by means of a Fourier transform infrared (FTIR) spectrometer (Shimadzu IRAffinity-1S, Columbia, MD, USA) in transmittance mode in the wavenumber range from 600 to 4000 cm^−1^ at a resolution of 4 cm^−1^. The inorganic matter content was quantified by calcination of PET samples in a muffle oven (Nabertherm B180, Lilienthal, Germany) in air at 625 °C for 4 h, following a heating ramp of 10 °C min^−1^. After calcination, the remaining ashes were weighed and also analyzed by FTIR in order to determine the nature of the inorganic matter present in the samples.

### 2.3. Catalytic Bio-Solvolysis of PET Waste

The catalytic bio-solvolysis reactions of PET waste were carried out in a 1000 mL three-necked glass reactor, equipped with a mechanical stirrer that was set at 250 rpm, a thermometer, and a reflux condenser. BioMEG as the glycolysis reagent and zinc acetate as the catalyst were added and heated to 185 °C. A PET/BioMEG molar ratio of 1:7.6 was used, with 1 wt.% of zinc acetate as a catalyst. Finally, 180 g of waste PET particles were fed into the reactor. Previous reaction conditions are optimum values determined in previous works carried out by authors to reach the full depolymerization of PET waste [9].

The reaction mechanism of solvolysis and influence of PET/BioMEG ratio were studied using PET-1 sample as feedstock. The reaction mechanism was determined by both HPLC (high-performance liquid chromatography) and NMR (nuclear magnetic resonance) methods. HPLC data were obtained by micrOTOF-QII (Bruker Daltonik Co., Ltd., Bremen, Germany) at a 200–1000 mg/kg concentration in tetrahydrofuran (THF). Details of the NMR analysis are explained below. Glycolyzed samples were taken at specific reaction times (0.5, 1, 2, 3 and 4 h) in order to quantify the degree of progress of the glycolysis reaction by monitoring the generation of the BHET monomer, dimers and oligomers. In addition, the influence of the PET/BioMEG molar ratio (1:2, 1:4, 1:6, 1:7.6) was also studied at a fixed reaction time of 2 h and a temperature of 185 °C in order to optimize one of the key variables in the bio-solvolysis processes.

Solvolysis reactions were also carried out with all the PET waste samples and at different reaction times (1, 2, 3 and 4 h) in order to study the PET conversion and BHET yield over time. The final reaction product, also known as glycolyzed, was filtered hot (100 °C) and under pressure (2 bar) in order to remove unreacted material. Then, demineralized water at a ratio of 2:1 vol. (Water/BioMEG) was added to the filtered product, and hot extraction of oligomers was carried out at 90 °C. Finally, the BHET monomer was obtained after a crystallization process at 5 °C and filtration using a 0.7 µm microfiber filter.

The results were evaluated in terms of PET conversion and BHET yield. PET conversion was gravimetrically determined on the basis of the weights of unreacted PET material using Equation (1).
(1)$${PET}_{conversion}(\%)=\frac{W_{PET}^{0}-W_{PET}^{t}}{W_{PET}^{0}}\times 100$$
where ${W\;}_{PET}^{0}$ (g) and ${W\;}_{PET}^{t}$ (g) refer to the initial weight of PET fed and the PET weight at a specific reaction time, respectively.

BHET yield was gravimetrically determined based on the obtained monomer weight after each run using Equation (2).
(2)$${BHET}_{yield}(\%)=\frac{W_{BHET}^{t}/{MW}_{BHET}}{W_{PET}^{0}/{MW}_{PET}}\times 100$$
where ${W\;}_{BHET}^{t}$ (g) and $W_{PET}^{0}$ (g) refer to the weight of BHET at a specific reaction time and the initial weight of PET, respectively. ${MW}_{BHET}$ and ${MW}_{PET}$ are the molecular weights of BHET (254 g mol^−1^) and PET monomer (192 g mol^−1^), respectively.

The overall process of solvolysis reaction is shown in Figure 3.

### 2.4. Purification of BHET Monomer

Since the obtained BHET monomer is coloured due to the pigments contained in the PET residue and also has other impurities such as metals, an exhaustive purification of the obtained monomer is carried out in order to remove the colour and reduce the metal content. For this purpose, a quantity of BHET monomer is dissolved in demineralized water (10 mL water per 1 g of BHET) and AC is added to this dissolution for the colour removal process (0.25 g AC per 1 g of BHET). After a given time, the AC is filtered out of the solution using a microfiber filter and a colourless BHET is obtained after crystallization. Finally, monomer rinsing is carried out using demineralized water, in order to reduce the concentration of metals in the BHET, especially Zn.

### 2.5. BHET Monomer Characterization

The chemical structure of the BHET samples was measured by means of a FTIR spectrometer (Shimadzu IRAffinity-1S) in transmittance mode in the wavenumber range from 600 to 4000 cm^−1^ at a resolution of 4 cm^−1^. DSC (TA Instruments DSC Q100 Model, New Castle, DE, USA) was used for thermal analysis of the BHET. Samples were heated from room temperature to 280 °C with a heating rate of 10 °C min^−1^ under a nitrogen flow of 50 mL min^−1^. Characterization of the colour of BHET produced after PET glycolysis was also carried out using a CM-2300d model spectrophotometer with a wavelength range between 360 nm and 740 nm. The CIE 1976 L*a*b* (CIELAB) colour space was used, which defines the colour of an object through three parameters: L* (100 = white; 0 = black), a* (positive = red; negative = green; 0 = gray) and b* (positive = yellow; negative = blue; 0 = gray). Proton nuclear magnetic resonance (^1^H-NMR) was also used to characterize the composition of the BHET samples. The proton NMR spectra were acquired in a 400 MHz Bruker apparatus. The sample preparation consisted of dissolving approximately 15 mg of the sample in 2 mL of deuterated chloroform. Under these conditions, all analyzed samples were completely soluble. Spectra were acquired using 16 accumulations for each spectrum. For the treatment of the spectra, the MestReNova program was used and all the spectra were referenced to the residual signal of the solvent, located at 7.26 ppm. Inductively coupled plasma-optical emission spectrometry (ICP-OES) analysis was carried out by Agilent ICP-OES 720 (Agilent Technologies, Santa Clara, CA, USA) in order to determine the Zn content of the obtained BHET samples.

## 3. Results and Discussion

### 3.1. Characterization of PET Waste

Table 2 summarizes the main properties of as-received PET waste samples prior to bio-glycolysis reaction. All PET waste samples show a similar moisture content with the exception of PET-3. The lower value of this sample can be related to the conditioning processes carried out in end-of-life tyre treatment facilities. In addition, PET-3 sample is the one with the highest inorganic matter content (4.77%) after calcination due to the presence of rubber and other inorganic materials in the textile fibres. A low bulk density value is observed for PET-4 sample (0.034 g cm^−3^), due to its textile nature, while the rest of the samples present typical density values for PET to be fed to a solvolysis process.

Figure 4a shows the FTIR spectrum of all PET wastes. All samples present the typical bands of PET [42], namely 1712 cm^−1^ (C=O stretching of the carboxylic acid group), 1410–1338 cm^−1^ (stretching of the C-O group, deformation of the O-H group and bending vibrational modes and oscillation of the ethylene glycol segment), 1240 cm^−1^ (terephthalate group, OOCC_6_H_4_-COO), 1090–1030 cm^−1^ (methylene group and vibrations of the C-O ester bond) and 720 cm^−1^ (interaction of polar ester groups and benzene rings).

As previously mentioned, the chemical nature of the inorganic compounds, present after the calcination of PET samples, was also characterized by FTIR. The following characteristic bands were identified (Figure 4b): silicates, in the 1000–1100 cm^−1^ range, which were accompanied by the band at 670 cm^−1^, showing the existence of silicon oxide compounds or silicates. Furthermore, bands of calcium carbonate were observed (1400 and 870 cm^−1^). These bands represent the union of CO_3_^2−^ associated with calcite [42]. The addition of calcium carbonate in PET plastics improves the thermal stability of the matrix [43].

### 3.2. Bio-Solvolysis of PET Waste and Reaction Mechanism Determination

The evolution of the BHET yield with different PET/BioMEG molar ratios was followed, for a fixed time (2 h), temperature (185 °C) and catalyst content (1 wt.%). PET-1 sample was used as feedstock for these tests. The BioMEG concentration range studied was between 1:2 and 1:7.6 (mol/mol) (Figure 5). The BHET yield significantly increases with the ratio until a stable equilibrium value is reached, around 48% for a PET/BioMEG ratio of 1:2 and above 86% for a PET/BioMEG ratio of 1:7.6. The low yield obtained for reactions with a PET/BioMEG ratio lower than 1:6 can be attributed to diffusional limitations since it is observed that the volume of BioMEG added is insufficient to mix adequately with the solid PET, resulting in mixtures of reduced homogeneity [44]. For this reason, and although the performance differences between the most highly diluted mixtures (1:6 and 1:7.6) are relatively small, the PET/BioMEG ratio of 1:7.6 has been selected as optimal for further studies, since higher ratios could lead to very diluted mixtures in which there is an (even higher) excess of glycol compared to the selected one.

Figure 6 shows the progress of the solvolysis reaction determined by means of HPLC and NMR analysis. PET-1 sample was used as feedstock for these tests. Composition of the glycolyzed samples taken at specific reaction times (0.5, 1, 2, 3 and 4 h) was analyzed in order to measure the extent of depolymerization by monitoring the generation of BHET monomer, dimers and oligomers. It should be noted that HPLC analysis excludes a solvent for the composition calculation. As shown in Figure 6a, all species analyzed by HPLC are formed in short reaction times and their concentration remains fairly constant over the progress of the reaction. The BHET monomer represents up to 84 wt.% of the solid content of the glycolyzed samples analyzed from short reaction times and remains constant throughout the depolymerization. Similarly, dimer and oligomer solid content of the analyzed samples remains constant throughout the reaction, reaching maximum values of 13.3 wt.% for the dimers and 1.8 wt.% for the oligomers. This indicates that equilibrium has been reached in the reaction and since the conversion of the dimer to the BHET monomer is a reversible process, polymerization can occur generating dimers or oligomers [45]. Therefore, extending the reaction after equilibrium has been reached causes the reaction to shift backwards, increasing the amount of dimer with a slight loss of the BHET monomer [46]. Figure 6b refers to the NMR results for glycolyzed samples extracted at different reaction times. It must be noted that NMR analysis includes solvent for the composition calculation. In this analysis, the species that can be found are BHET, ethylene glycol, dimer and trimer. In this series the main product is ethylene glycol, since it is the chemical agent used in the glycolysis of PET. At the beginning of the glycolysis reaction (0.5 h sample), this product represents up to 73.2 wt.% of the whole mixture and in the last stages of the reaction (4 h sample), the percentage of this compound slightly decreases to 60.5 wt.%. Then, the second major compound is BHET monomer and its behavior is the opposite; initially, the value is 24.3 wt.% and then its content increases to 34.6 wt.%. This maximum value is reached after 2 h so the BHET content in the samples has reached a plateau value. A similar upward trend is observed for the dimer and trimer species of the BHET, the maximum values being around 5.3 wt.% and 1.0 wt.%, respectively. In addition, it should be noted that a ratio between the dimer/trimer content has been detected during the NMR analysis of the glycolyzed samples, being the content in moles of dimer about 10 to 12 times higher than the trimer content. This ratio could probably be determined by the solubility of the products in the reaction medium, considering that the higher molecular weight species were not soluble in the samples and therefore were not extracted in the analyzed aliquots. The trends previously described and observed by the NMR analytical technique determine that as the glycolysis reaction time increases, dimers and low-molecular-weight oligomers are also generated by the polymerization process occurring due to the reversibility of the depolymerization reaction of PET into BHET [47].

Figure 7 shows the evolution of a PET conversion and BHET yield over reaction time for all wastes studied. As seen in Figure 7a, it can be concluded that all PET samples reach almost full depolymerization after 2 h of bio-solvolysis reaction, maintaining a constant conversion profile from this value onward, with the exception of PET-3 sample because its high content of rubber and other textile fibres causes the overall conversion of the sample into monomer to decrease to values lower than 12%. The optimum reaction time for obtaining maximum values of both PET conversion and BHET yield can be set at 2 h, reaching 97.1% PET conversion and 76.3% BHET yield in the case of PET-2 sample. From Figure 7b it can be concluded that extending the reaction after equilibrium has been reached causes the reaction to shift backwards, increasing the amount of dimer with a slight loss of the BHET monomer. Nevertheless, high conversion and yield values have been obtained during bio-solvolysis of PET waste, the best results being for the monolayer materials and somewhat lower for the multilayer and fibre containing samples, including PET-3 which is the most difficult waste stream to treat via glycolysis [48].

### 3.3. BHET Purification and Characterization

The obtention of purified BHET was confirmed by FTIR analysis. As can be seen in Figure 8a, the spectra of all analyzed samples match with that of the BHET monomer. Glycolysis of PET leads to the formation of hydroxyl groups, reflected in the band at 3451 cm^−1^, present at the spectra of all samples [49]. The intense absorption band corresponding to the ester group at 1712 cm^−1^ was also detected, whereas the bands between 3100 and 2800 cm^−1^ are attributed to the aliphatic groups (-CH_2−_). Other characteristic absorption peaks were observed at 2879 cm^−1^ and 2952 cm^−1^ (C-H bonds), 1716 cm^−1^ (C=O bond), 1240 cm^−1^ (C(=O)-O), 1097 cm^−1^ (O-C-C) and peaks at 1504, 1411, 728 cm^−1^ for the C-H bonds of the aromatic ring. DSC thermograms of the obtained BHET samples can be seen in Figure 8b, corresponding to a heating scan from room temperature to 280 °C at a heating rate of 10 °C min^−1^. All thermograms show the endothermic peak corresponding to the melting temperature at around 110 °C, matching that found for BHET in the literature [50], thus meaning that for all studied wastes the product obtained is mainly purified BHET.

Table 3 shows the chemical composition of unpurified and purified BHET samples, determined by means of NMR analysis. The quantification of the composition was carried out before and after the BHET purification process in order to determine the performance of this step. For the unpurified samples (Figure 9a), the majority of the product is BHET with a composition greater than 94% in mol. After that, the BHET dimer has a variable percentage, although in all cases this value is less than 5% in mol. This product could come from a partial glycolysis of the PET chain or a condensation reaction between two molecules of BHET [51]. Regarding ethylene glycol, the presence of this compound is the lowest detected in all samples with a content lower than 2% in mol. The molecular weight of all analyzed samples is higher than pure BHET (254.24 g/mol). This is due to the fact that there is a higher proportion of dimer than ethylene glycol, as the dimer content increases the overall molecular weight of the sample, whereas an increase in ethylene glycol would cause the opposite effect. For the purified samples (Figure 9b), the proposed purification process has succeeded in triggering the BHET content, exceeding 98% in mol for all the samples. For the specific cases of monomers coming from the bio-solvolysis of PET-3, PET-4 and PET-5 waste samples, the BHET content of the samples reached 99.5% in mol. The dimer content of the purified samples is lower compared to the unpurified ones and can be set at less than 1.5%. Furthermore, no signal of glycolysis solvent was found in the analyzed samples, so the traces of free ethylene glycol were completely removed. Since no glycol was detected and the content of the dimer species was very low, the molecular weight values of the monomer samples are very close to that of the theoretical BHET value. Nevertheless, all the measured values were slightly higher than the theoretical one because only the dimer exists as a by-product. The NMR spectra of a purified BHET sample and its characteristic peaks are shown in Figure 10. The spectrum shows an intense peak in the aromatic hydrogen region at 8.11 ppm corresponding to the protons of the terephthalate ring (labeled with the letter a). Additionally, the pseudo-triplet peaks positioned at 4.49 and 3.79 come from the hydrogens of the methylene groups identified as b and c, respectively, related to ethylene glycol molecules esterified at one end with the terephthalate ring. Finally, the peak located at 4.70 ppm corresponds to the species referred to as BHET dimer and trimer. The presence of both species can be discerned in that peak, since at 4.70 ppm the dimer is found, while at 4.69 ppm the trimer is located. To know the composition between both products, it is necessary to perform a deconvolution of that peak and to know the areas of each peak. In the analyzed samples, it has been observed that the peak is deconvoluted with only one peak centered at 4.70 ppm; therefore, only the dimer species is detected. In summary, the compounds that can be detected in the analyzed samples are BHET, dimer and free ethylene glycol. From this, it is possible to know the composition of the samples, in addition to the average molecular weight (Mn) of the whole mixture considering the area of the peaks and the molecular weights of each species. Mn parameter can be used to measure the purity of the sample, since the closer it is to the theoretical value of the BHET (254.24 g/mol) the purer the product will be. It should be noted that dimer and trimer species will cause an increase in Mn of the mixture, while the presence of ethylene glycol will have the opposite effect.

Table 4 shows the measured colour parameters and Zn content of the produced BHET samples. In PET chemical recycling processes, the coloured and metallic impurities have to be removed from the BHET to produce good-quality rPET. Despite the intense colour of the unpurified BHET samples, after purification processes involving active carbon, filtration and washing with demineralized water, the final product loses much of its colour and clean and purified BHET is obtained. The whiteness (L*) of all samples greatly increased from around 60 to over 95 (L* = 100 for pure white), indicating an improvement in the crystallinity of the samples [52]. Furthermore, the Zn content of the unpurified samples was significantly reduced under the investigated purification conditions, reaching values lower than 5 mg/kg, which is indeed a clear advantage for the subsequent polymerization of BHET to produce colourless plastics, which are especially demanded in the food sector.

## 4. Conclusions

The chemical recycling of barely studied complex PET wastes, such as multilayer trays, coloured bottles and postconsumer textiles, by means of catalytic glycolysis assisted by a biobased solvent, was examined. Purified BHET monomer (up to 99.5% in mol) was obtained from those complex materials after a bio-solvolysis reaction. A good compromise between PET conversion (91.3–97.1%) and BHET yield (71.5–76.3%) was established after 2 h of reaction, under the following operating conditions: PET/BioMEG molar ratio = 1:7.6, 1 wt.% of zinc acetate as catalyst, T = 185 °C and stirring rate = 250 rpm. However, NMR analysis pointed out that, as the reaction time increases, dimers and low molecular weight oligomers are also generated by the polymerization process occurring due to the reversibility of the depolymerization reaction of PET into BHET.

The developed monomer purification process using activated carbon was successful in removing the colour and reducing the metal content of the BHET samples. By applying this purification strategy, the whiteness (L*) of BHET greatly increased from around 60 to over 95 (L* = 100 for pure white) and the Zn content was significantly reduced from around 200 to 2 mg/kg. The high purity of the BHET samples was demonstrated via NMR analysis with a monomer content of up to 99.5% in mol.

The characterization of the BHET samples obtained through the bio-solvolysis of PET waste indicates a clear advantage of the developed process compared to other chemical recycling processes under development, because it allows the subsequent polymerization of BHET to produce colourless plastics that are suitable for food contact applications. Therefore, it has been demonstrated that the biobased ethylene glycol achieves the same PET conversion and BHET yield values as a fossil ethylene glycol, also taking into account that the inclusion of a biobased glycol reduces the environmental impacts of chemical recycling.

## Figures and Tables

**Figure 1 polymers-16-00142-f001:**
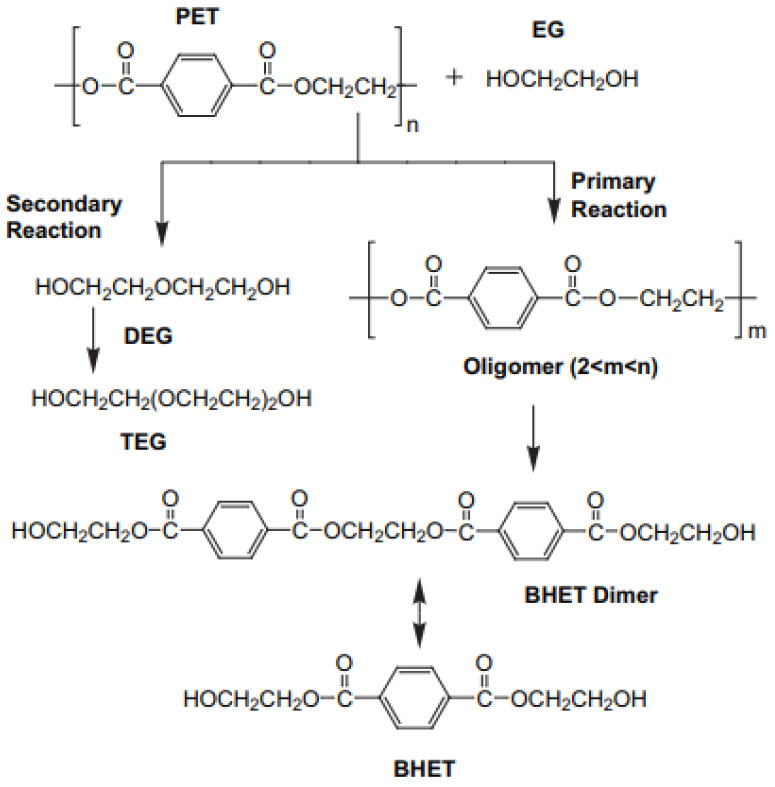
Mechanism for glycolysis reaction [7].

**Figure 2 polymers-16-00142-f002:**
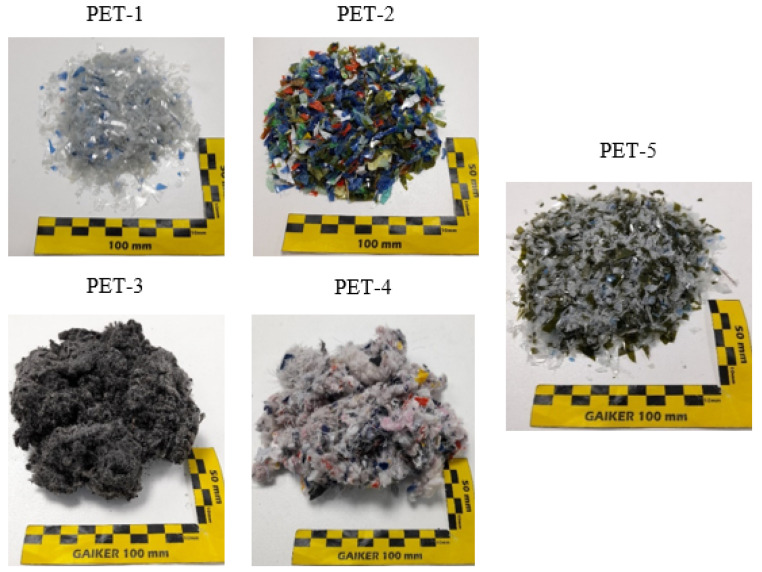
Conditioned complex PET waste samples.

**Figure 3 polymers-16-00142-f003:**
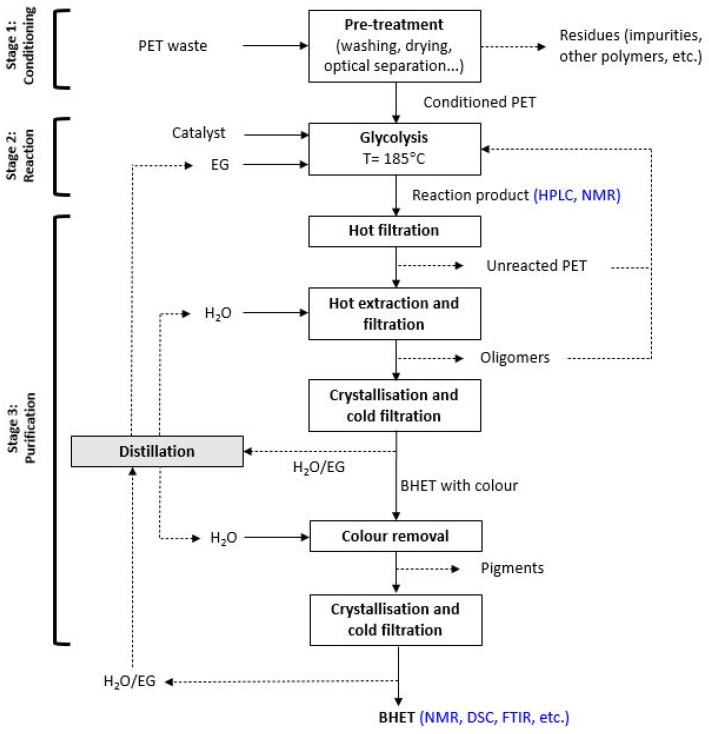
PET glycolysis reaction and product purification process.

**Figure 4 polymers-16-00142-f004:**
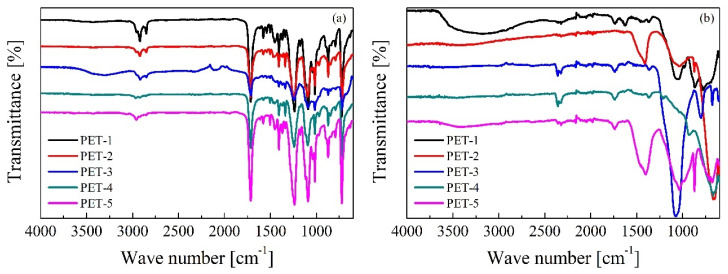
FTIR spectrum of (**a**) PET waste samples and (**b**) their ashes.

**Figure 5 polymers-16-00142-f005:**
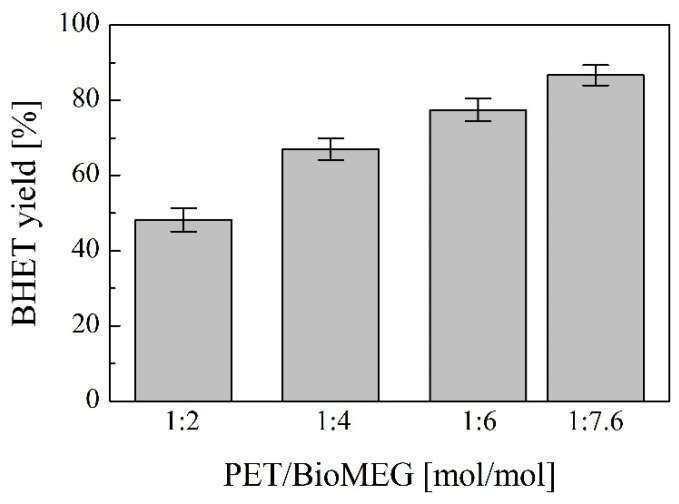
Evolution of BHET yield versus PET/BioMEG molar ratio. Operating conditions: temperature = 185 °C, time= 2 h, stirring rate = 250 rpm and catalyst content = 1 wt.%. Feedstock: PET-1.

**Figure 6 polymers-16-00142-f006:**
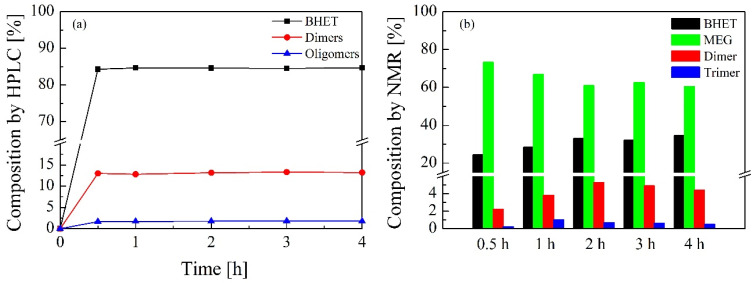
Progress of the solvolysis reaction by means of (**a**) HPLC and (**b**) NMR. Operating conditions: temperature = 185 °C, PET/BioMEG = 1/7.6 (mol/mol), stirring rate = 250 rpm and catalyst content = 1 wt.%. Feedstock: PET-1.

**Figure 7 polymers-16-00142-f007:**
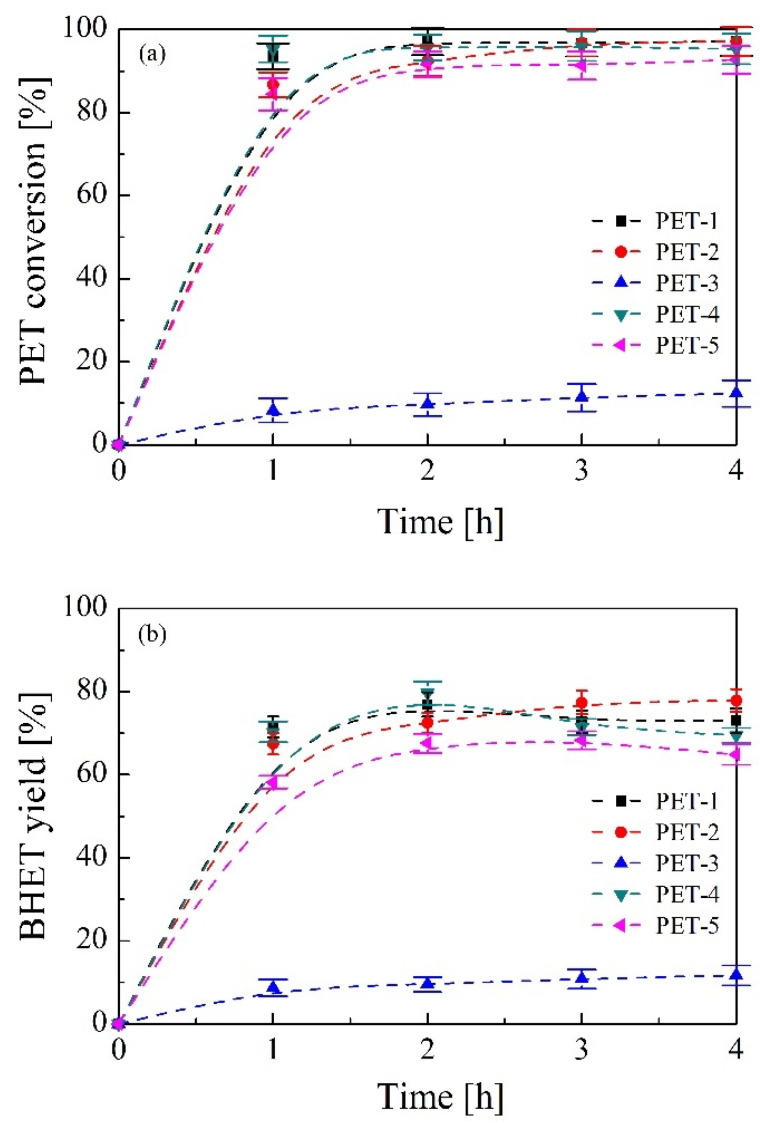
Effect of reaction time on (**a**) PET conversion and (**b**) BHET yield. Operating conditions: temperature = 185 °C, PET/BioMEG = 1/7.6 (mol/mol), stirring rate= 250 rpm and catalyst content = 1 wt.%.

**Figure 8 polymers-16-00142-f008:**
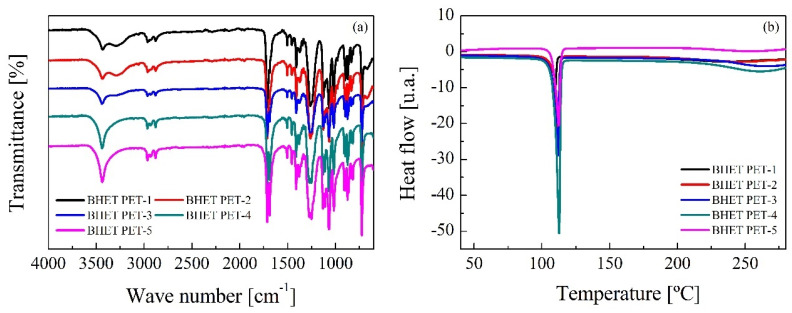
Characterization of BHET samples by means of (**a**) FTIR and (**b**) DSC. Operating conditions: temperature = 185 °C, time = 4 h, PET/BioMEG = 1/7.6 (mol/mol), stirring rate = 250 rpm and catalyst content = 1 wt.%.

**Figure 9 polymers-16-00142-f009:**
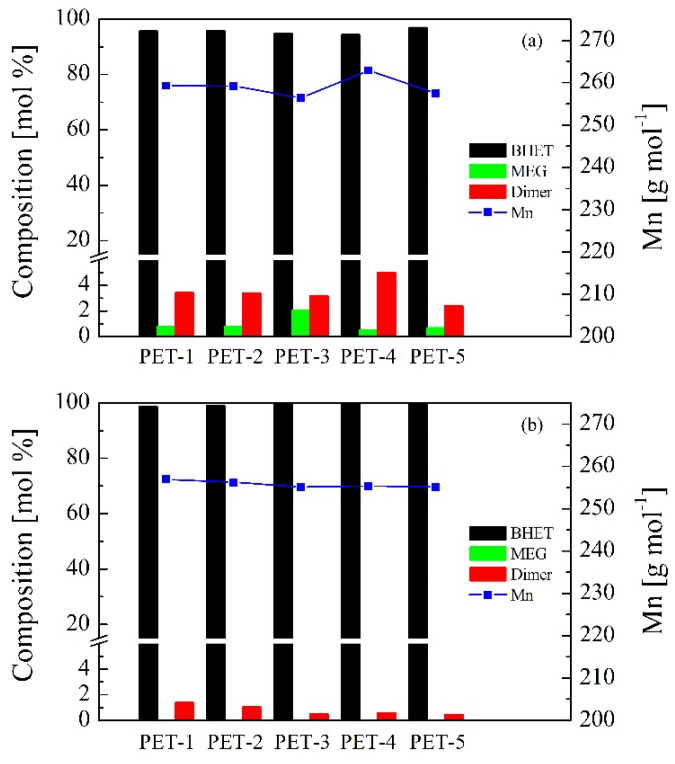
Molar composition of (**a**) unpurified and (**b**) purified BHET samples by NMR. Operating conditions: temperature = 185 °C, time = 4 h, PET/BioMEG = 1/7.6 (mol/mol), stirring rate = 250 rpm and catalyst content = 1 wt.%.

**Figure 10 polymers-16-00142-f010:**
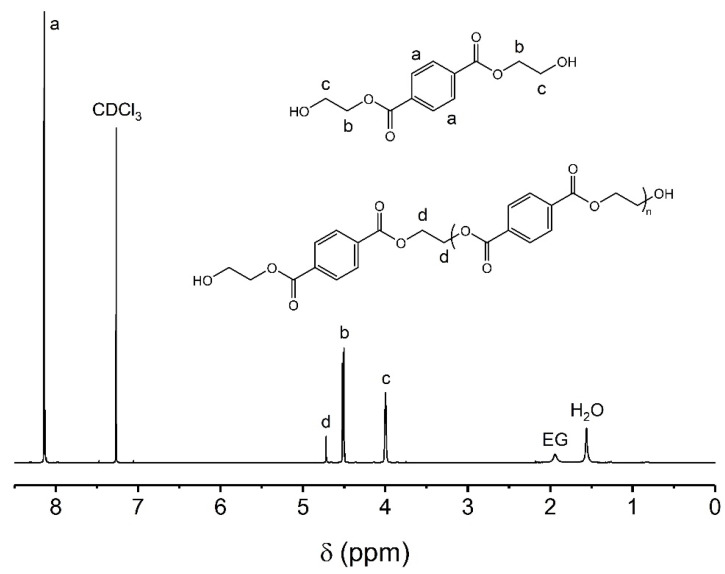
NMR spectra of a purified BHET sample. Operating conditions: temperature = 185 °C, time = 4 h, PET/BioMEG = 1/7.6 (mol/mol), stirring rate = 250 rpm and catalyst content = 1 wt.%. Feedstock: PET-1.

**Table 1 polymers-16-00142-t001:** PET waste samples characteristics.

PET	Material Description	Complexity
PET-1	Postconsumer monolayer trays	Low polyolefin concentration
PET-2	Postconsumer coloured bottles	Highly coloured plastic
PET-3	Textile from end-of-life tyres	Low density, presence of rubber
PET-4	Textile from postconsumer clothing	Low density, presence of other fibres
PET-5	Postconsumer multilayer trays	High polyolefin concentration

**Table 2 polymers-16-00142-t002:** PET waste samples characterization results.

PET	Moisture (%)	Ash Content (%)	Bulk Density (g cm^−3^)
PET-1	2.42	0.10	0.462
PET-2	2.35	0.34	0.220
PET-3	0.76	4.77	0.261
PET-4	2.11	1.08	0.034
PET-5	2.42	0.06	0.394

**Table 3 polymers-16-00142-t003:** BHET sample composition (unpurified/purified) according to NMR analysis.

PET	BHET (mol%)	MEG (mol%)	Dimer (mol%)	Trimer (mol%)	Mn (g/mol)
PET-1	95.74/98.61	0.81/0.00	3.45/1.39	0.00/0.00	259.30/256.91
PET-2	95.82/98.94	0.80/0.00	3.38/1.06	0.00/0.00	259.21/256.28
PET-3	94.79/99.52	2.04/0.00	3.17/0.48	0.00/0.00	256.40/255.16
PET-4	94.48/99.41	0.51/0.00	5.00/0.59	0.00/0.00	262.87/255.37
PET-5	96.93/99.53	0.69/0.00	2.38/0.47	0.00/0.00	257.47/255.14

**Table 4 polymers-16-00142-t004:** CIELAB colour parameters and Zn content of obtained BHET samples.

Sample	L*	a*	b*	Zn (mg/kg)
BHET PET-1 unpurified	88.19	−0.63	4.68	182.3
BHET PET-1 purified	95.23	−0.12	1.43	1.9
BHET PET-2 unpurified	88.39	−1.10	7.36	185.7
BHET PET-2 purified	95.68	−0.02	1.12	2.5
BHET PET-3 unpurified	62.71	6.66	17.52	201.4
BHET PET-3 purified	91.44	−0.23	2.32	4.3
BHET PET-4 unpurified	64.33	9.37	6.24	193.6
BHET PET-4 purified	93.33	−0.13	1.39	3.6
BHET PET-5 unpurified	86.37	−2.30	17.55	200.9
BHET PET-5 purified	93.80	−0.18	1.52	2.7

## Data Availability

The data presented in this study are available on request from the corresponding author.

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
