# Peer review of "Improving the Sustainability of Catalytic Glycolysis of Complex PET Waste through Bio-Solvolysis"

_polymers, 2024, doi:10.3390/polym16010142_

Round 1
Reviewer 1 Report
Comments and Suggestions for Authors
The article entitled “Improving the sustainability of catalytic glycolysis of complex PET waste through bio-solvolysis” is devoted to polyethylene terephthalate (PET) waste recycling into bis(2-hydroxyethyl) terephthalate (BHET) widely used in the synthesis of polymers. The article is of current interest because polymer waste recycling contributes to environmental problem solving. PET glycolysis into BHET is a well-known method of PET recycling and zinc acetate is a conventional catalyst for PET chemical recycling. Therefore, the authors need to clearly emphasize the novelty and uniqueness of their work in the Introduction section. Moreover, other questions and comments are below.
1. Previously published research works on the glycolysis of PET into BHET using zinc acetate as a catalyst should be mentioned in the Introduction section.
2. What is the difference (besides the origin) between biobased ethylene glycol and ethylene glycol obtained by conventional methods?
3. Is it possible to prevent the formation of polymerization products during glycolysis?
4. Low quality of Figure 1.
Comments on the Quality of English Language-
Reviewer 2 Report
Comments and Suggestions for Authors
This paper endeavors to elucidate the treatment of complex polyethylene terephthalate (PET) waste employing a biobased monoethylene glycol (BioMEG) as a depolymerization agent to enhance the sustainability of the chemical recycling process. While the reaction itself is not novel, certain aspects remain unclear and are expounded upon in the ensuing discussion.
1. Five kinds of PET were utilized in this study, and the captions for Figures 5, 6, 9, and 10 should include the respective reaction conditions. Additionally, it is imperative to specify the types of PET employed in each reaction.
2. The discrepancy in BHET yield between Figures 6a and Figure 7 requires clarification.
3. The conclusion posits, "A favorable compromise between PET conversion (91.3-97.1%) and BHET yield (71.5-76.3%) was achieved after a 2-hour reaction period, employing the following conditions: PET/BioMEG molar ratio = 1:7.6, 1 wt.% of zinc acetate as a catalyst, T = 185 ºC, and stirring rate = 250 rpm." However, the author omits discussion on the rationale behind selecting zinc acetate, temperature, and stirring rate as optimal conditions.
4. The author should provide the specifications of BioMEG in the experimental section, and if its properties align with those of MEG, clarification is needed on the innovation aspect.
5. In the experimental section, it is essential to quantify and present all raw materials, including specifying what AC is and detailing the amount of AC used for decolorization.
6. The author should elucidate the calculation method for the dimer obtained from NMR data.
7. In Figure 5, consider changing the PET/EG mole ratio to a mass ratio, aligning with current practices in the field.
8. The statement "the PET/BioMEG ratio of 1:7.6 has been selected as optimal" contradicts Figure 5, where it is described as maximal. The author should address why alternative ratios, such as 1:9 or 1:10, were not explored to determine the true optimal ratio.
9. In Figure 6b, an explanation is needed for the summation of BHET+dimer+trimer not equating to 100%.

none
Reviewer 3 Report
Comments and Suggestions for Authors
Dear all,
please, improve your paper with all report that you can find in several papers.
Se my comments inside your PDF.

Round 2
Reviewer 2 Report
Comments and Suggestions for Authors
It can be accepted.